# Nutritional status of Indian adolescents (15-19 years) from National Family Health Surveys 3 and 4: Revised estimates using WHO 2007 Growth reference

Madhavi Bhargava[1,2]*, Anurag Bhargava[2,3], Sudeep D. Ghate[4], R. Shyama Prasad Rao[4]

1 Department of Community Medicine, Yenepoya Medical College, Mangalore, Karnataka, India, 2 Center for Nutrition Studies, Yenepoya (Deemed to be University), Mangalore, Karnataka, India, 3 Department of General Medicine, Yenepoya Medical College, Mangalore, Karnataka, India, 4 Yenepoya Research Centre, Yenepoya (Deemed to be University), Mangalore, Karnataka, India

* madhavibhargava4@gmail.com

## Abstract

**Data Availability Statement:** All relevant data are within the paper and its Supporting Information files.

### Background

The National Family Health Surveys (NFHS) in India apply adult cutoffs of nutritional status for the estimation of undernutrition/overweight in the 15–19 age group. The prevalence of thinness in boys and girls thus estimated is 58.1% and 46.8% in NFHS-3, and 45% and 42% in NFHS-4 respectively. But the WHO recommends using age and sex-specific reference for adolescents. We reanalyzed the nutritional status of the adolescents using the WHO 2007 Growth Reference to obtain revised estimates of thinness, overweight and stunting across states, rural-urban residence, and wealth quintiles.

### Methods and findings

Demographic information, anthropometric data, and wealth index were accessed from the Demographic and Health Survey (DHS) database. We re-analyzed the anthropometric data using WHO AnthroPlus software which uses the WHO 2007 Growth reference. The revised estimates of thinness assessed by BMI-for-age z-scores in boys and girls was 22.3% (95% CI: 21.6, 23.0) and 9.9% (95%CI: 9.5, 10.3) in NFHS-3 and 16.5% (95%CI: 16.0,17.0) and 9% (95%CI: 8.9, 9.2) in NFHS-4 respectively. Stunting was found to be 32.2% (95% CI: 31.6, 32.9) in boys and 34.4% (95% CI: 34.2, 34.7) in girls in NFHS-4. This was higher than that in NFHS-3; 25.2% (95% CI: 24.4, 26) in boys and 31.2 (95% CI: 30.6, 31.8) in girls. There was a clear socioeconomic gradient as there were higher thinness and stunting in rural areas. There was wide variation among the states with pockets of a double burden of malnutrition.

**Funding:** MB received grant from The United Nations Children's Fund (UNICEF) SSFA/2019/07. URL of the website available at: https://unicef.in/Karnataka. The funders did not play any role in the design, analysis and preparation of the manuscript.

**Competing interests:** The authors have declared that no competing interests exist.

## Conclusion

Using the adult cutoffs significantly overestimates thinness in adolescents in the age group of 15–19 years old in India. Stunting, which is an indicator of long term nutrition is also widely prevalent in them. Future editions of DHS and NFHS should consider adolescents as a separate age group for nutritional assessment for a better understanding of nutritional transition in the population.

## Introduction

The World Health Organization (WHO) defines adolescence as 10–19 years [1]. It is a key decade in the life course with implications on adult health, socio-economic well-being of a country and even the health of the future children. Adolescents comprise 16% of the total world population [2]. Asia has more than half of the world's adolescents while according to the Census 2011, 20% of India's population are adolescents [3,4].

Adolescence is a period of rapid growth and development, second only to infancy, with dramatic biological, psychological changes often shaped by socio-cultural factors. It is usually divided into two phases: early adolescence (10–14 years) and late adolescence (15–19 years) [5]. Physiologically, the early years are dominated by pubertal changes and the later stages by sexual maturation and development of adult roles and responsibilities. The nutritional issues in this age group have commonalities with children and adults with some added dimensions of puberty, psychological changes, and growth spurt which are crucial for current, future and intergeneration health [6]. Poor nutrition, on the one hand, can lead to delay or failure in achieving maturation with a stunted linear growth perpetuating the cycle of poverty and intergenerational undernutrition. On the other hand, there is an increased risk of non-communicable diseases (NCDs) [7]. The nutritional transition that is occurring in some low-middle income countries is resulting in a double burden of overweight and obesity in some population groups, along with the existing high proportion of undernutrition in others [8–10].

The National Family Health Surveys (NFHS)are periodic health and demographic surveys that include adult men and women in the age group of 15–54 years and 15–49 respectively and the adolescents (15–19 years) are included in them [11,12].The NFHS reports have so far used the adult cutoffs for nutritional assessment for this age group; i.e., those with a BMI <18.5 kg/m$^2$ are considered thin, those between 25–29.9 kg/m$^2$ are overweight, and BMI >30kg/m$^2$ are considered obese [11–13]. The proportions thus classified as thin were 58.1% and 46.8% in NFHS-3,and 45% and 42% in NFHS-4 in boys and girls respectively [11,12]. However, their nutritional assessment should be done using the WHO 2007 Growth reference curves [14].

We re-analyze the nutritional status of adolescents in the age group of 15–19 years(referred to as adolescents hereafter) included in NFHS-3 and 4 using the age and sex-specific WHO 2007 Growth reference to obtain revised estimates of thinness, overweight and stunting in boys and girls and compare them with currently reported estimates. We also compare these estimates across urban and rural populations, wealth quintiles and explore sub-national heterogeneity and inequalities based on socio-economic class.

## Methods

### Data source—National Family Health Survey

The NFHS is conducted at regular intervals in India since 1992–93 under the stewardship of the Ministry of Health and Family Welfare and coordinated by the International Institute of

Population Sciences (IIPS), Mumbai. The NFHS-3 was conducted in 2005–6 and NFHS-4 in 2015–16. Data of both these surveys were extracted in Microsoft Excel from the Demographic and Health Surveys (DHS) database [15].

## Study population and sample size

The NFHS-3 covered a representative sample of 109,041 households from the Census of 2001and included adult men and women in the age group of 15–54 years and15-49 years respectively. The number of boys and girls in the age group of 15–19 years was 26,086 and 26,238 respectively, making upa total of 52,324 participants.

TheNFHS-4 covered a representative sample of 601,509 households from the Census of 2011 with adult men and women in the same age categories as NFHS-3. A total of 277,059 adolescents in the age group of 15–19 years: 142,162 boys and 134,897 girls were included from these households.

## Anthropometry

Trained field staff visited the households and any adult member capable of providing information served as the respondent for the 'Household Questionnaire'. Other members were consulted only if necessary and this was followed by a 'Biomarker or Measurement Questionnaire'. This included among other things, the measurement of weight and height using standard techniques for the available members in the under-five and 15-49-year-old adults. In the NFHS-3, weight was measured using a solar-powered electronic SECA scale with a digital screen (manufactured under the guidance of UNICEF) and measuring board designed by Shorr Productions for use in survey settings [11]. In the NFHS-4, digital weighing machine (SECA 874) to the nearest of one gram and stadiometer (SECA 213) to the nearest of 0.1cm were used. This was done by trained field staff and monitored by field supervisors. The percentage of missing anthropometric information was 6% in the under-five participants, 12.2% in men (15–54) and 5.7% in women (15–49) in NFHS-4 [12], but corresponding information for participants in the NFHS-3 is not available.

We used the WHO 2007 Growth Reference to assess the nutritional status of adolescents (14). Age, sex, weight, and height recorded in the NFHS-3 and NFHS-4 surveys, were accessed from DHS database, extracted in Microsoft Excel, and imported as a TXT file into the 'nutritional survey tool' of the WHO AnthroPlus software (v1.04) [16]. This is specifically used for children and adolescents in the age group of 6–19 years. Height-for-age z-scores (HAZ) and BMI-for-age z-scores (BAZ) were used to identify stunting and thinness (and overweight/obesity) respectively. The nutritional survey tool of the software generates 'standard reports' and gives the prevalence of nutritional abnormalities as percentages and confidence intervals (CI), stratified by sex along with mean z-scores and standard deviations (SD). The software uses default lower and upper SD boundary as flag limits to identify any extreme or potentially incorrect z-scores. These are -6SD and +6SD for HAZ scores and -5SD and +5SD for BAZ scores and any values beyond these limits get automatically removed.

## Interpretation of anthropometric data

All the height-for-age z-scores <-2 SD compared to the WHO median were considered as stunting; while BAZ scores < -2SD and <-3SD defined thinness and severe thinness respectively [14]. The BAZ score at +1SD for 19 years coincides with the adult BMI of 25 kg/m$^2$; which is the cut-off for overweight in adults [13]. As a result, BAZ >+1SD was used to classify overweight and >+2SD for obesity [14].

## Statistical analyses

We used descriptive statistics to describe stunting, thinness, and overweight and stratified these by sex, urban-rural residence, and wealth quintiles. An independent t-test was used to compare z-scores available as a continuous variable. ANOVA was used to test the means across the five wealth quintiles. A two-sided p-value of $<0.05$ was considered as statistically significant. The statistical analysis was done using StataCorp 2009 (Stata Statistical Software: Release 11. College Station, TX: StataCorp LP).

## Data visualization

Z-score plots for boys and girls in comparison with the WHO reference curves were generated using the AnthroPlus Software (v 1.0.4). Representation of sub-national heterogeneity in nutritional status was done using R-Software in the form of maps and comparative bar-charts.

## Ethics statement

The respondents in the NFHS undergo an informed consent process for participation in the survey after approval of the protocol by the institutional review board of the IIPS. These NFHS datasets are available for download from the DHS program after registration [15]. This study was a secondary data analysis of de-identified data; therefore ethics committee approval was not obtained.

## Results

Table 1 describes select demographic characteristics of the adolescent participants in NFHS-3 and 4. The NFHS-3 included 52,324 and NFHS-4 included 277,059 adolescents. There was more urban representation (23,802, 45.5%) in NFHS-3 as compared to NFHS-4 (76,095, 27.5%). The poorer and poorest quintiles together contributed 26% in NFHS-3 and 45% in NFHS-4.

**Table 1. Demographic characteristics of adolescents in (15–19 years) enumerated in NFHS-3 (2005–6) and NFHS-4 (2015–16).**

| Demographic Characteristic | | NFHS-3 N = 52,324 (%) | NFHS-4 N = 277,059 (%) |
|---|---|---|---|
| **Age** | 15 years | 9,897 (18.9) | 59,300 (21.4) |
| | 16 years | 11,093 (21.2) | 56,000 (20.2) |
| | 17 years | 9,599 (18.3) | 52,196 (18.8) |
| | 18 years | 13,311 (25.4) | 63,574 (22.9) |
| | 19 years | 8,424 (16.1) | 45,989 (16.6) |
| **Gender** | Males | 26,086 (49.9) | 142,162 (51.3) |
| | Females | 26,238 (50.1) | 134,897 (48.7) |
| **Residence** | Urban | 23,802 (45.5) | 76,095 (27.5) |
| | Rural | 28,522 (54.5) | 200,964 (72.5) |
| | Do not know/missing | 8 (0.1) | 417 (0.2) |
| **Wealth Index** | Poorest | 5,807 (11.1) | 58,883 (21.3) |
| | Poorer | 7,803 (14.9) | 66,055 (23.8) |
| | Middle | 10,915 (20.9) | 59,769 (21.6) |
| | Richer | 13,207 (25.2) | 50,112 (18.1) |
| | Richest | 14,592 (27.9) | 42,240 (15.2) |

NFHS = National Family Health Survey.

## Nutritional status of adolescents (15–19 years) in NFHS-3 and NFHS-4 using WHO 2007 Growth reference

Data available for final analysis in the adolescents who underwent anthropometry and after removal of improbable values by AnthroPlus Software was 68% in NFHS-3 (48.4% boys; 87.4% girls) and 52% in NFHS-4 (13.4% boys; 92.9% girls).

Table 2 describes the mean BAZ and HAZ scores for the adolescents in NFHS-3 stratified by the urban-rural residence and the wealth quintiles. In the NFHS-3, the mean BAZ and HAZ scores were significantly lower in the rural adolescents and those in the poorer quintiles, and the mean HAZ score was significantly lower in the girls. The mean BAZ for boys was -1.17 (SD: 1.14) and that for girls was -0.71 (SD: 1.01) (p<0.001); mean HAZ score for girls was -1.58 (SD: 0.89) which was -1.63 (SD: 0.88) in rural girls. In NFHS-4 (Table 3), the mean BAZ for the boys was -0.85 (SD: 1.2) and that in girls was -0.64 (SD: 1.03) (p<0.001). Stunting was consistently higher in the girls as compared to the boys (p<0.001) in all sub-groups; being highest in rural girls (mean HAZ -1.69; SD: 0.9) and lowest in urban boys (mean HAZ -1.45,

**Table 2. Nutritional status of adolescents (15–19 years) subjected to anthropometry in NFHS-3 (N = 35,570).**

| Population | Sex | N | Mean BAZ (SD) | Mean HAZ (SD) |
|---|---|---|---|---|
| **India** | Overall | 35,570 | -0.87 (1.08) | -1.51 (0.93) |
| | Boys* | 12,635 | -1.17 (1.14) | -1.38 (0.99) |
| | Girls* | 22,935 | -0.71 (1.01) | -1.58 (0.89) |
| **Residence** | | | | |
| **Urban†** | Overall | 16,146 | -0.81 (1.16) | -1.42 (0.95) |
| | Boys* | 6,231 | -1.07 (1.22) | -1.27 (0.99) |
| | Girls* | 9,915 | -0.65 (1.09) | -1.51 (0.90) |
| **Rural†** | Overall | 19,424 | -0.93 (1.01) | -1.58 (0.92) |
| | Boys* | 6,404 | -1.26 (1.06) | -1.48 (0.97) |
| | Girls* | 13,020 | -0.76 (0.94) | -1.63 (0.88) |
| **Wealth Index^** | | | | |
| **Poorest** | Overall | 3,764 | -1.06 (0.94) | -1.82 (0.91) |
| | Boys | 1,181 | -1.42 (1.0) | -1.72 (0.98) |
| | Girls | 2,583 | -0.89 (0.86) | -1.87 (0.87) |
| **Poorer** | Overall | 5,311 | -1.01 (0.97) | -1.73 (0.9) |
| | Boys | 1,846 | -1.34 (1.01) | -1.65 (0.96) |
| | Girls | 3,465 | -0.83 (0.9) | -1.77 (0.86) |
| **Middle** | Overall | 7,580 | -0.94 (1.03) | -1.62 (0.88) |
| | Boys | 2,717 | -1.27 (1.07) | -1.54 (0.94) |
| | Girls | 4,863 | -0.76 (0.97) | -1.66 (0.85) |
| **Richer** | Overall | 9,067 | -0.89 (1.07) | -1.49 (0.91) |
| | Boys | 3,255 | -1.19 (1.11) | -1.37 (0.95) |
| | Girls | 5,812 | 0.72 (1.01) | -1.56 (0.87) |
| **Richest** | Overall | 9,848 | -0.66 (1.19) | -1.2 (0.93) |
| | Boys | 3,636 | -0.90 (1.27) | -1.01 (0.96) |
| | Girls | 6,212 | -0.52 (1.12) | -1.31 (0.90) |

N = numbers with plausible anthropometric measurements available; BAZ = BMI-for-age z-score; HAZ = Height-for-age z-score; The BMI for age and Height for age z scores are based on the 2007 WHO Growth Reference. Independent t-test for BAZ and HAZ

\*boys and girls: p < 0.001

†urban and rural: p<0.001

^ ANOVA for BAZ and HAZ for wealth quintiles: p<0.001.

Table 3. Nutritional status of adolescents (15–19 years) subjected to anthropometry in NFHS-4 (N = 144,320).

| Population | Sex | N | Mean BAZ (SD) | Mean HAZ (SD) |
|---|---|---|---|---|
| **India** | Overall | 144,320 | -0.67 (1.06) | -1.64 (0.93) |
| | Boys* | 18,970 | -0.85 (1.2) | -1.54 (1.06) |
| | Girls* | 125,350 | -0.64 (1.03) | -1.66 (0.9) |
| **Residence** | | | | |
| **Urban†** | Overall | 38,674 | -0.55 (1.15) | -1.54 (0.93) |
| | Boys* | 5,695 | -0.7 (1.29) | -1.45 (1.06) |
| | Girls* | 32,979 | -0.52 (1.12) | -1.56 (0.9) |
| **Rural†** | Overall | 105,646 | -0.71 (1.02) | -1.68 (0.92) |
| | Boys* | 13,275 | -0.91 (1.15) | -1.57 (1.05) |
| | Girls* | 92,371 | -0.69 (0.99) | -1.69 (0.9) |
| **Wealth India^** | | | | |
| **Poorest** | Overall | 30,314 | -0.82 (0.94) | -1.94 (0.88) |
| | Boys | 3,446 | -1.09 (1.08) | -1.92 (1.0) |
| | Girls | 26,868 | -0.78 (0.92) | -1.94 (0.87) |
| **Poorer** | Overall | 34,714 | -0.74 (0.99) | -1.74 (0.9) |
| | Boys | 4,348 | -0.97 (1.10) | -1.68 (1.01) |
| | Girls | 30,366 | -0.71 (0.97) | -1.75 (0.88) |
| **Middle** | Overall | 31,682 | -0.67 (1.06) | -1.61 (0.90) |
| | Boys | 4,204 | -0.88 (1.18) | -1.52 (1.02) |
| | Girls | 27,478 | -0.64 (1.03) | -1.62 (0.88) |
| **Richer** | Overall | 26,549 | -0.60 (1.12) | -1.47 (0.91) |
| | Boys | 3,752 | -0.77 (1.23) | -1.33 (1.04) |
| | Girls | 22,797 | -0.57 (1.09) | -1.49 (0.89) |
| **Richest** | Overall | 21,059 | -0.43 (1.18) | -1.31 (0.93) |
| | Boys | 3,220 | -0.48 (1.34) | -1.18 (1.09) |
| | Girls | 17,839 | -0.42 (1.15) | -1.33 (0.90) |

n = numbers with plausible anthropometric measurements available; BAZ = BMI-for-age z-score; HAZ = Height-for-age z-score; The BAZ and HAZ are based on the 2007 WHO Growth Reference. Independent t-test for BAZ and HAZ

*boys and girls: $p < 0.001$

†urban and rural: $p < 0.001$

^ ANOVA for BAZ and HAZ for wealth quintiles: $p < 0.001$.

SD: 1.06). The adolescents in the poorest quintile performed worse than their counterparts in the NFHS-3, the overall mean HAZ was -1.94 (SD: 0.88) and that in girls was -1.94 (SD: 0.87).

Figs 1A–4B show the BAZ and HAZ curves of the urban and rural participants of both sexes compared to the WHO reference. There is a visible shift of curves to the left in both the NFHS.

## Comparison of currently reported nutritional status of the adolescents in NFHS-3&4 with the revised estimates

Table 4 compares the burden of thinness in adolescents as reported by the NFHS-3 & 4 with the revised estimates using the WHO 2007 Growth reference. Using adult cut-offs for thinness this was 58.1% in boys and 46.8% in girls in NFHS-3. Using the WHO 2007 Growth reference, it was 22.3% (95%CI: 21.6, 23.0) in boys and 9.9% (95%CI: 9.5, 10.3) in girls. Similarly, NFHS-4 reported 44.8% in boys and 41.9% thinness in girls, which reduce to 16.5% (95%CI: 16.0,17.0) and 9% (95%CI: 8.9, 9.2) in boys and girls respectively.

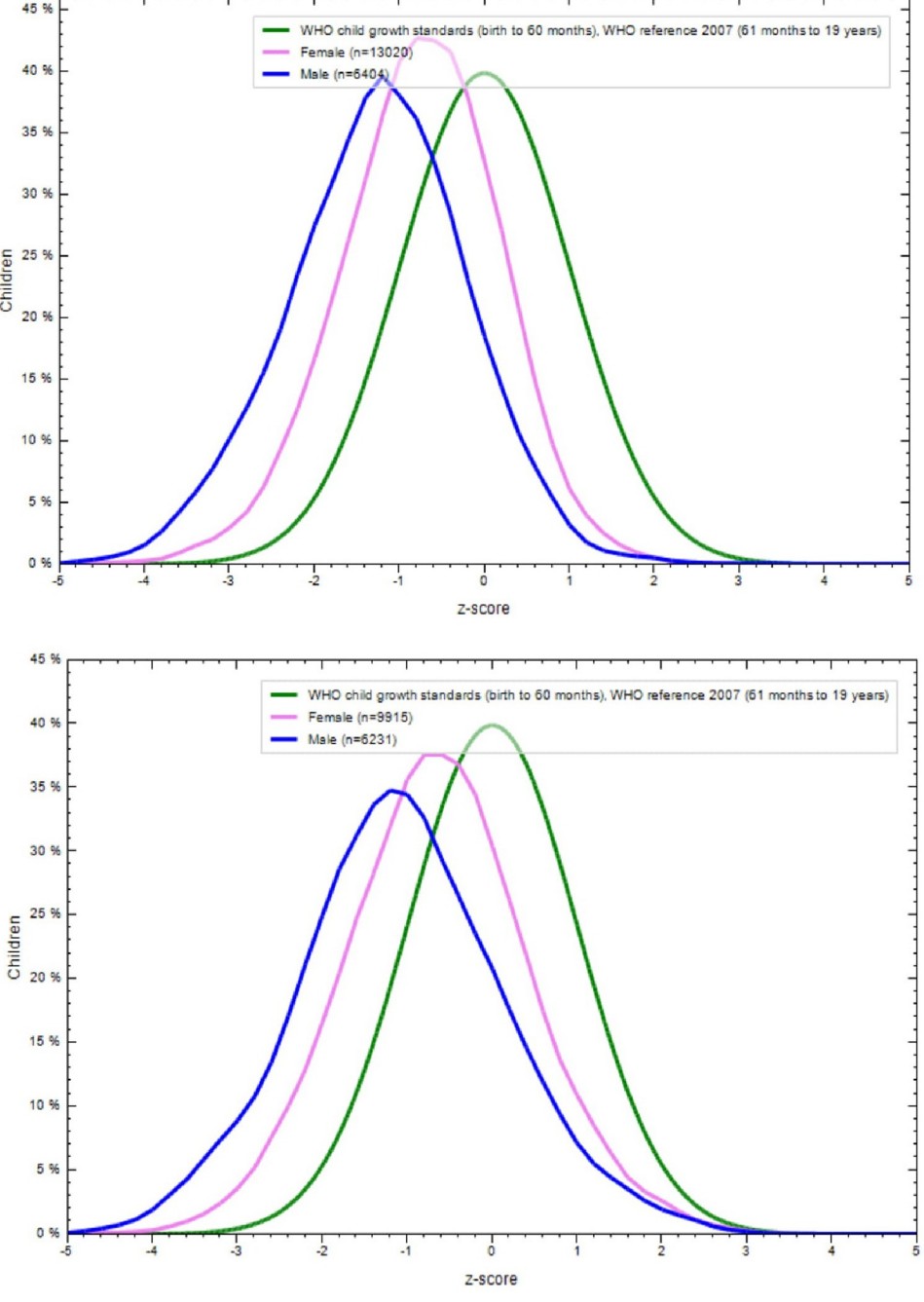

**Fig 1.** A. BMI-for-age z-scores of rural adolescents (15–19 years), NFHS-3 (2005–6). B. BMI-for-age z-scores of urban adolescents (15–19 years), NFHS-3 (2005–6).

According to the NFHS reports the prevalence of short-statured women of reproductive age-group ($< 145$ cm) was 11.7% in NFHS-3 and 12.7% in NFHS-4. In the NFHS, data is available for stunting in the under-fives, but not for stunting in adolescents or adults. We found stunting to be 32.2% (95% CI: 31.6, 32.9) in boys and 34.4% (95% CI: 34.2, 34.7) in girls in the NFHS-4. This was higher than the corresponding prevalence in the NFHS-3; 25.2% (95% CI: 24.4, 26) in boys and 31.2% (95% CI: 30.6, 31.8). A significant increase in stunting from

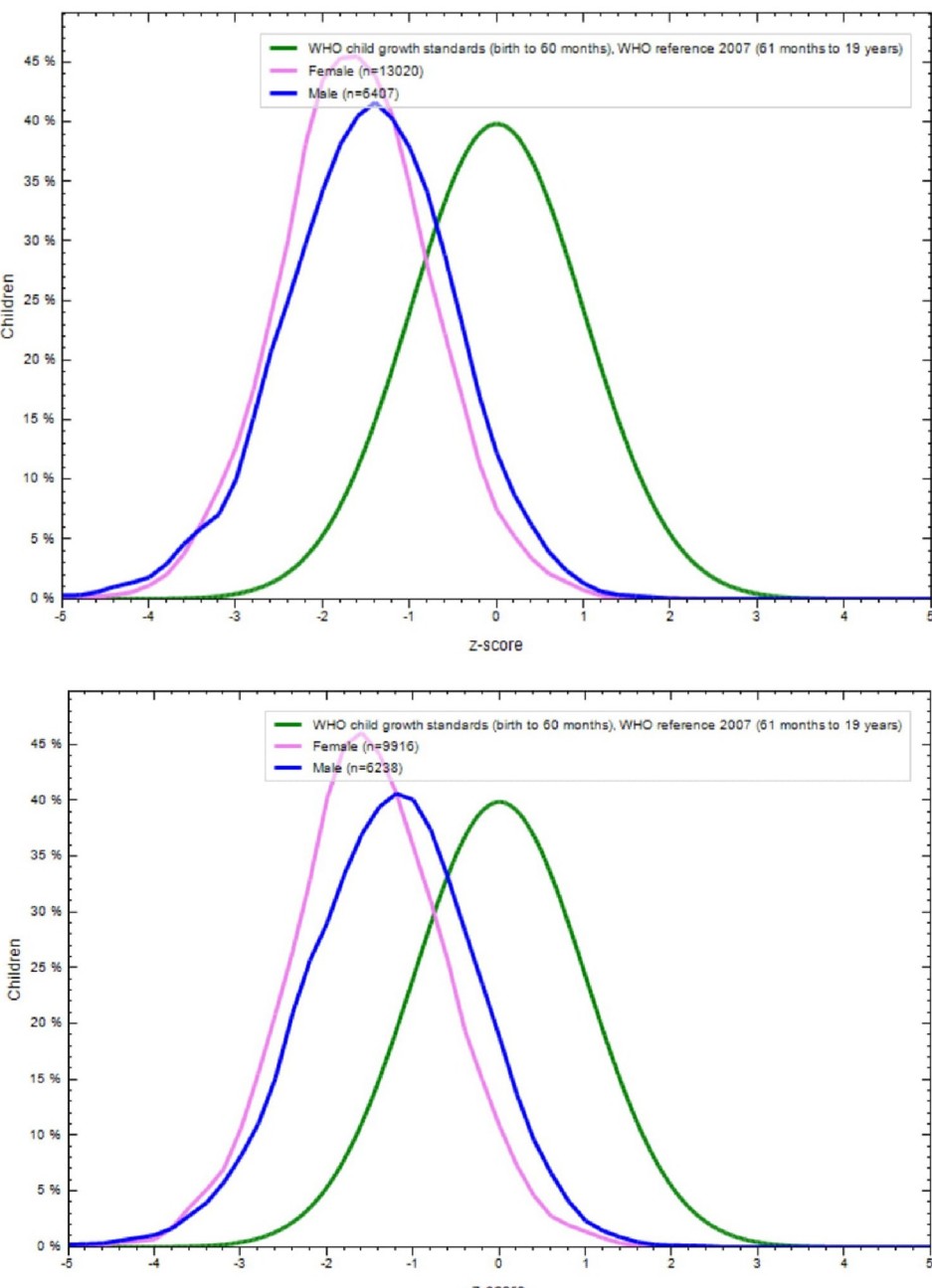

**Fig 2.** A. Height-for-age z-scores of rural adolescents (15–19 years), NFHS-3 (2005–6). B. Height-for-age z-scores of urban adolescents (15–19 years), NFHS-3 (2005–6).

NFHS-3 to 4 for both sexes, urban/rural and within the wealth quintiles (p<0.001) was observed.

Table 5 compares the over-nutrition (overweight/obesity) in adolescents as reported currently in NFHS-3 and 4 with the revised estimates using WHO 2007 Growth reference. Overweight has been reported as 1.7%in boys and 2.4% in girls in the NFHS-3 which increased to3% (95% CI: 2.7, 3.3) and 4.3% (95% CI: 4, 4.6) respectively using WHO 2007 Growth reference. In NFHS-4, the same was 4.8%in boys and 4.2% in girls which increased to 6.2% (95% CI: 5.9, 6.4) and 5% (95% CI: 4.8, 5.1) respectively in the revised estimates.

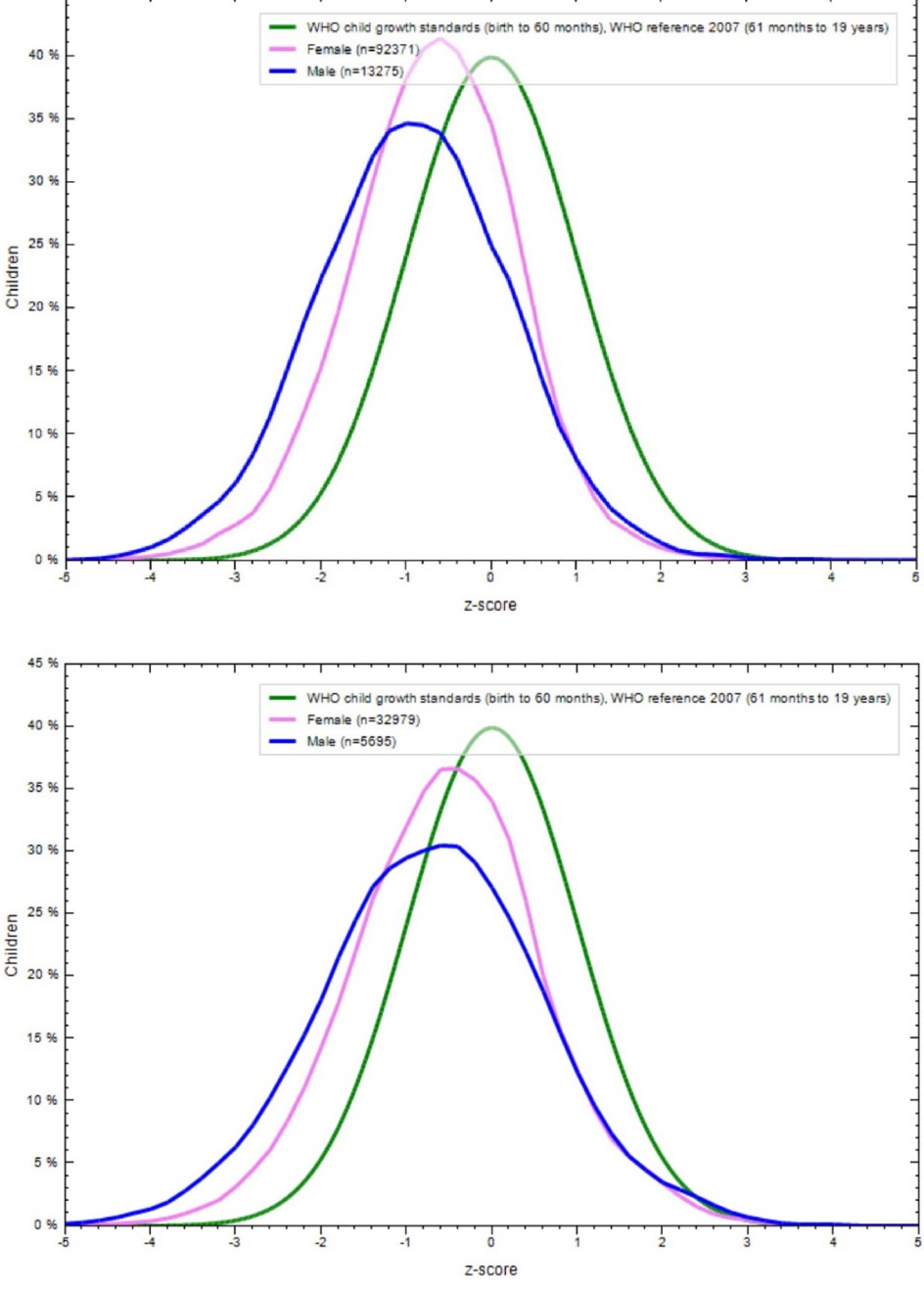

**Fig 3.** A. BMI-for-age z-scores of rural adolescents (15–19 years), NFHS-4 (2015–16). B. BMI-for-age z-scores of urban adolescents (15–19 years), NFHS-4 (2015–16).

## Sub-national revised estimates of thinness, overweight in thinness and stunting

Tables 6 and 7 describe thinness in adolescents across Indian states in NFHS-3 and 4. The overall thinness in NFHS-3 was 14.3% (95%CI: 13.9, 14.7). The state of Maharashtra had the highest levels of thinness: 24.9% (95% CI: 23.3, 26.5). It was the least in all the north-eastern states, Delhi, Jammu and Kashmir, and Punjab. In NFHS-4, the overall thinness was 10% (95%

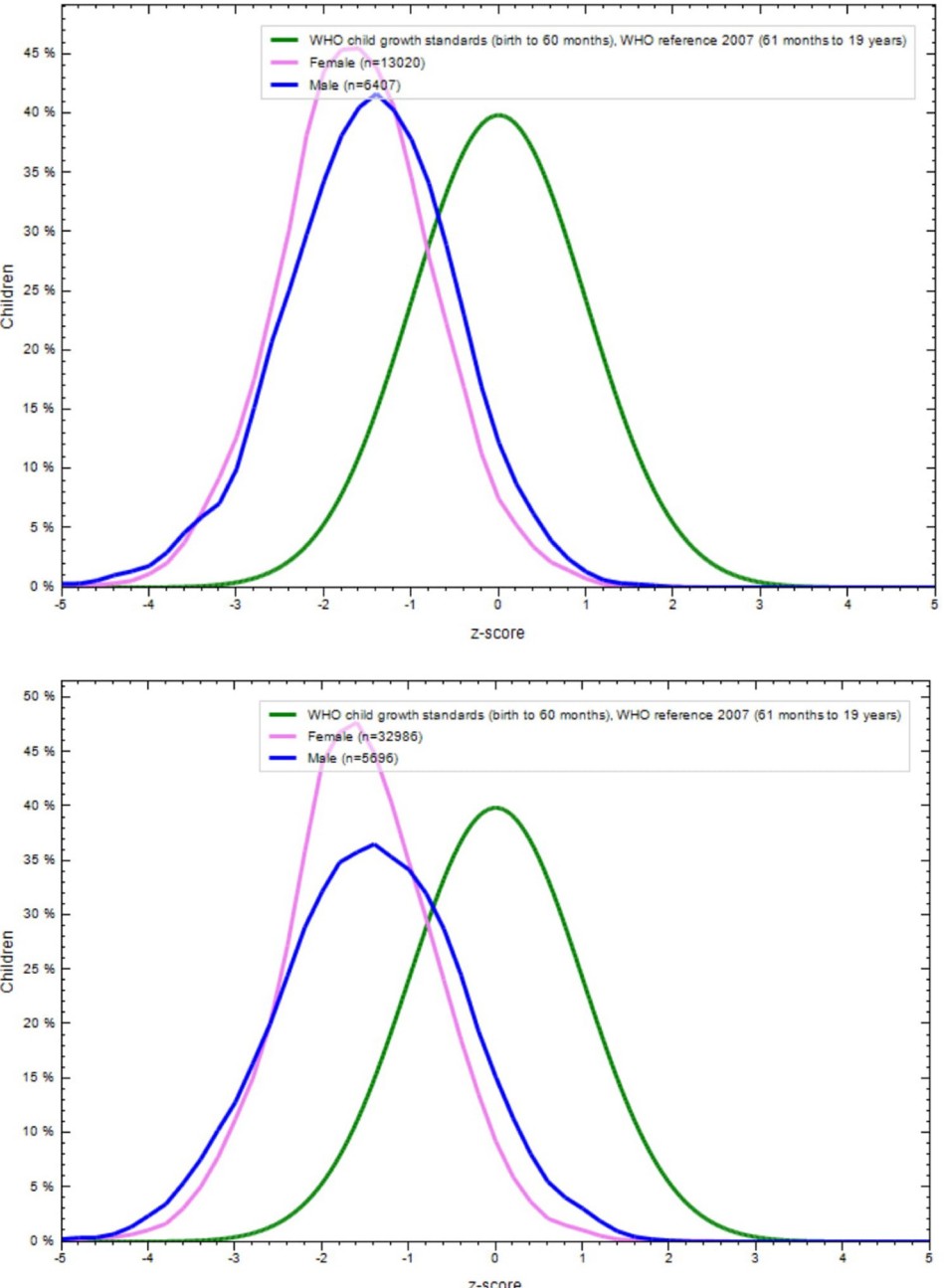

**Fig 4.** A. Height-for-age z-score of rural adolescents (15–19 years) in NFHS-4 (2015–16). B. Height-for-age z-score of urban adolescents (15–19 years) in NFHS-4 (2015–16).

CI: 9.8. 10.1) and it was most pronounced in Gujarat: 19.3% (95%CI: 18.2, 20.5), Telangana (17.2; 95%CI: 15, 19.4), and Maharashtra (15.4; 95% CI: 14.4, 16.3) and again more in the boys. The Northeastern states had the least thinness. Thinness in less than 10% of adolescents was prevalent in 8 states in NFHS-3 and 20 states in the NFHS-4

The overall prevalence of overweight was 3.8% (95%CI: 3.6, 4) in NFHS-3 and 5.1 (95%CI: 4.9, 5.2) in NFHS-4. There were 15 states with overweight greater than the national average (highest in Punjab and Kerala) in the NFHS-3 while there were 29 states and union territories

**Table 4. Prevalence of undernutrition in adolescents (15–19 years) in NFHS– 3 and 4 using adult cut offs vs. that based on 2007 WHO Growth Reference.**

| NFHS | Based on WHO cutoffs for adults [#] | | Based on WHO 2007 Growth Reference for adolescents | | | |
|---|---|---|---|---|---|---|
| | Thinness | Severe thinness^ | Thinness% (CI) | Severe thinness^ % (CI) | Stunting %(CI) | Severe stunting^% (CI) |
| **NFHS-3 (2005–6)** | | | | | | |
| Both Sexes(35,570) | NA* | NA* | 14.3(13.9, 14.7) | 2.9(2.8, 3.1) | 29.1(28.6, 29.6) | 5.3(5, 5.5) |
| Boys(12,635) | 58.1 | 29.3 | 22.3(21.6, 23) | 5.8(5.4, 6.2) | 25.2(24.4, 26) | 5.2(4.8, 5.6) |
| Girls(22,935) | 46.8 | 20.9 | 9.9(9.5, 10.3) | 1.4(1.2, 1.5) | 31.2(30.6, 31.8) | 5.3(5, 5.6) |
| **NFHS-4 (20015–16)** | | | | | | |
| Both Sexes(144,320) | NA* | NA* | 10(9.8, 10.1) | 1.7(1.6, 1.8) | 34.1 (33.9, 34.4) | 6.5(6.4, 6.6) |
| Boys(18,970) | 44.8 | 23.0 | 16.5(16, 17) | 3.6(3.3, 3.8) | 32.2(31.6, 32.9) | 8.6 (8.2, 9) |
| Girls(125,350) | 41.9 | 23.2 | 9(8.8, 9.2) | 1.4(1.4, 1.5) | 34.4(34.2, 34.7) | 6.2(6.1, 6.3) |

NFHS = National Family Health Survey; BMI = Body Mass Index.

*Combined figures of both sexes are not officially available in any NFHS report for this age group

[#]indicators for adults are based on the recommendations by the 1995 WHO Expert Committee (14)

^ Percent severe thinness, percent severe stunting, included in the overall figures of thinness and stunting

According to WHO 2007 growth reference, thinness and severe thinness is BMI-for-age z-scores <-2SD and <-3SD respectively

According to WHO 2007 growth reference stunting and severe stunting is Height-for-age z-scores < -2SD and -3SD.

(UTs) with overweight more than the national average (highest in Chandigarh and Kerala) in the NFHS-4. None of the states had a prevalence of overweight/obesity of more than 10%in the former whereas there were 7 such states and UTs in the latter.

Fig 5 describes the sub-national distribution of thinness and overweight in NFHS-3 and 4. State-level stunting in the NFHS-3 is described in Table 8 and Fig 6. The national prevalence of stunting in adolescents was 29.1% (95% CI: 28.6. 29.6) in NFHS-3 with range of 63.9% (95%CI: 60, 67.9)in Meghalaya to 14.6% (95%CI: 12.3%, 17%) in Punjab. Fifteen states had stunting greater than the national prevalence. Apart from the Northeastern states, these included Jharkhand (43.9%), Bihar (37.6%), West Bengal (36.8%), Orissa (34.8%), Chhattisgarh (32.2%), Uttar Pradesh (29.9%) and Delhi (29.8%). Six states had a prevalence of stunting of less than 20%.

**Table 5. Prevalence of overweight/obesity in adolescents (15–19 years) in NFHS-3 and 4 using adult cutoffs vs. those for adolescents based on 2007 WHO Growth reference.**

| NFHS | Adult cut-offs[#] | | Based on WHO 2007 Growth Reference | |
|---|---|---|---|---|
| | BMI≥25Overweight | BMI ≥30.0Obese^ | Overweight% (CI) | Obese^% (CI) |
| **NFHS-3 (2005–6)** | | | | |
| Both Sexes(35,570) | NA* | NA* | 3.8 (3.6, 4) | 0.6 (0.5, 0.6) |
| Boys (12,635) | 1.7 | 0.2 | 3 (2.7, 3.3) | 0.6 (0.5, 0.7) |
| Girls (22,935) | 2.4 | 0.2 | 4.3 (4, 4.6) | 0.6 (0.5, 0.7) |
| **NFHS-4 (2015–16)** | | | | |
| Both Sexes(144,320) | NA* | NA* | 5.1 (4.9, 5.2) | 0.8 (0.8, 0.9) |
| Boys (18,970) | 4.8 | 1.0 | 6.2 (5.9, 6.4) | 1.2 (1, 1.3) |
| Girls (125,350) | 4.2 | 0.8 | 4.9 (4.8, 5.1) | 0.8 (0.7, 0.8) |

NFHS = National Family Health Survey; BMI = Body Mass Index; BMI = Body Mass Index; BAZ = BMI-for-age z-scores.

*Combined figures of both sexes are not officially available in any NFHS report for this age group

[#]indicators for adults are based on the recommendations by the 1995 WHO Expert Committee (14)

^ Obesity percentage is included in overweight.

According to WHO 2007 Growth Reference, overweight is BMI-for-age z-score >+1SD and obesity is <+2SD.

**Table 6. State level prevalence of thinness (BMI-for-age z-scores <-2SD) and overweight (BMI-for-age z-scores >1 SD) in NFHS-3 (2005–6).**

| Region (N) | % Girls with thinness (CI) | % of boys with thinness (CI) | % thinness both sexes (CI) | % Girls with overweight (CI) | % boys with overweight (CI) | % overweight both sexes (CI) |
|---|---|---|---|---|---|---|
| Andhra Pradesh (2,384) | 12.4 (10.5, 14.2) | 24.6 (22, 27.2) | 18 (16.5, 19.6) | 0.9 (0.4, 1.5) | 4.4 (3.2, 5.7) | 5.8 (4.8, 6.7) |
| Arunachal (558) | 2.7 (1, 4.4) | 4.5 (0.9, 8.2) | 3.2 (1.7, 4.8) | 3.7 (1.7, 5.7) | 1.3 (0, 3.4) | 3 (1.5, 4.6) |
| Assam (886) | 8.9 (6.6, 11.1) | 18.9 (13.6, 24.1) | 11.5 (9.4, 13.7) | 3.2 (1.8, 4.6) | 2.6 (0.3, 4.8) | 3 (1.9, 4.2) |
| Bihar (1,070) | 10.9 (8.7, 13.1) | 27 (21.1, 32.9) | 14.5 (12.3, 16.6) | 1.9 (0.9, 2.9) | 0.4 (0, 1.5) | 1.6 (0.8, 2.4) |
| Chhattisgarh (1,053) | 11.8 (9.5, 14.1) | 27.8 (22.1, 33.4) | 15.8 (13.5, 18) | 1.5 (0.6, 2.4) | 0.4 (0, 1.3) | 1.2 (0.5, 1.9) |
| Delhi (680) | 7.2 (4.8, 9.7) | 7.6 (3.8, 11.4) | 7.4 (5.3, 9.4) | 6.4 (4.1, 8.7) | 5.2 (2, 8.4) | 6 (4.2, 7.9) |
| Goa (648) | 16.5 (13.1, 19.8) | 27.3 (19.9, 34.8) | 19 (15.9, 22.1) | 6 (3.8, 8.2) | 4.7 (1, 8.4) | 5.7 (3.8, 7.6) |
| Gujarat (910) | 17.2 (14.3, 20.2) | 30.8 (24.7, 36.9) | 20.8 (18.1, 23.5) | 3.9 (2.3, 5.4) | 1.7 (0, 3.5) | 3.3 (2.1, 4.5) |
| Haryana (759) | 8.8 (6.3, 11.3) | 23 (17.1, 28.9) | 12.8 (10.3, 15.2) | 3.5 (1.9, 5.1) | 6.6 (3, 10.1) | 4.3 (2.8, 5.9) |
| Himachal (730) | 16.6 (13.3, 19.9) | 25.2 (19, 31.5) | 19 (16.1, 21.9) | 3 (1.5, 4.6) | 2.5 (0.1, 4.9) | 2.9 (1.6, 4.1) |
| Jammu and Kashmir (897) | 5.7 (3.9, 7.6) | 17.5 (12.6, 22.4) | 9 (7.1, 11) | 6 (4.1, 8) | 2.8 (0.6, 5) | 5.1 (3.6, 6.6) |
| Jharkhand (768) | 8.9 (6.5, 11.3) | 20 (14, 26) | 11.6 (9.3, 13.9) | 1.7 (0.6, 2.9) | 1.1 (0, 2.8) | 1.6 (0.6, 2.5) |
| Karnataka (1,810) | 14.5 (12.3, 16.7) | 36.7 (33.3, 40.2) | 24 (22%, 26%) | 3.4 (2.2, 4.5) | 2.6 (1.4, 3.8) | 3 (2.2, 3.9) |
| Kerala (705) | 7.3 (5, 9.6) | 26.8 (19.8, 33.8) | 11.9 (9.5, 14.4) | 8.2 (5.8, 10.6) | 1.8 (0, 4.1) | 6.7 (4.8, 8.6) |
| Manipur (1,381) | 2.4 (1.2, 3.5) | 6.1 (4.1, 8.1) | 4.1 (3, 5.1) | 5.3 (3.6, 6.9) | 2.6 (1.2, 3.9) | 4.1 (3, 5.1) |
| Meghalaya (596) | 2.7 (1.1, 4.3) | 5.3 (1.4, 9.2) | 3.4 (1.8, 4.9) | 1.6 (0.3, 2.8) | 4 (0.5, 7.4) | 2.2 (0.9, 3.4) |
| Maharashtra (2,856) | 15.7 (13.8, 17.6) | 34.5 (31.9, 37) | 24.9 (23.3, 26.5) | 4.4 (3.3, 5.5) | 3.9 (2.8, 4.9) | 4.1 (3.4, 4.9) |
| Mizoram (426) | 1 (0, 2.2) | 4.3 (0.2, 8.4) | 1.9 (0.5, 3.3) | 6.8 (3.8, 9.8) | 3.4 (0, 7.1) | 5.9 (3.5, 8.2) |
| Madhya Pradesh (1,787) | 12.5 (10.6, 14.3) | 24.5 (20.7, 28.4) | 15.8 (14.1, 17.6) | 3.4 (2.4, 4.4) | 2.4 (1, 3.9) | 3.1 (2.3, 4) |
| Nagaland (1,515) | 2.3 (1.2, 3.4) | 7.6 (5.6, 9.5) | 4.8 (3.7, 5.9) | 3.7 (2.3, 5.1) | 2.6 (1.4, 3.8) | 3.2 (2.3, 4.1) |
| Orissa (6,251) | 9.3 (7.3, 11.4) | 20.2 (14.8, 25.5) | 11.7 (9.7, 13.7) | 3.2 (2, 4.4) | 3 (0.6, 5.4) | 3.2 (2.1, 4.2) |
| Punjab (888) | 7.8 (5.6, 9.9) | 17.4 (12.6, 22.3) | 10.6 (8.5, 12.7) | 6.7 (4.6, 8.7) | 8.1 (4.6, 11.7) | 7.1 (5.3, 8.8) |
| Rajasthan (1,135) | 11.2 (9, 13.4) | 23.4 (18.4, 28.4) | 14.4 (12.3, 16.4) | 2.4 (1.3, 3.5) | 0.3 (0, 1.2) | 1.9 (1, 2.7) |
| Sikkim (594) | 2.9 (1.2, 4.5) | 4.9 (1, 8.7) | 3.4 (1.8, 4.9) | 7.1 (4.6, 9.6) | 5.6 (1.5, 9.6) | 6.7 (4.6, 8.8) |
| Tamil Nadu (1,656) | 13.5 (11.2, 15.8) | 29.7 (26.5, 33) | 21.1 (19.1, 23.1) | 7.1 (5.3, 8.8) | 4.9 (3.3, 6.4) | 6 (4.9, 7.2) |
| Tripura (566) | 10.8 (7.8, 13.8) | 28.7 (20.3, 37.1) | 14.7 (11.7, 17.7) | 4.1 (2.1, 6) | 0.8 (0, 2.8) | 3.4 (1.8, 4.9) |
| Uttarakhand (778) | 9.5 (7, 11.9) | 21.6 (15.2, 28) | 12.2 (9.8, 14.6) | 4.2 (2.5, 5.8) | 0 (0, 0.3) | 3.2 (1.9, 4.5) |
| Uttar Pradesh (4,822) | 7.5 (6.5, 8.6) | 19.8 (18.2, 21.5) | 13.5 (12.5, 14.5) | 3.6 (2.8, 4.4) | 2.2 (1.6, 2.8) | 2.9 (2.4, 3.4) |
| West Bengal (1,632) | 10.3 (8.6, 12.1) | 17.7 (13.8, 21.7) | 12.1 (10.5, 13.7) | 4.9 (3.7, 6.1) | 2.9 (1.1, 4.7) | 4.4 (3.4, 5.4) |

Thinness, severe thinness, overweight in adolescents are based on the 2007 WHO Growth Reference. Overweight percentages include obesity.

State-level stunting in the NFHS-4 is described in Table 9 and Fig 6. The national prevalence was 34.1% (95%CI: 33.9, 34.4). Meghalaya continued to top the list but with a marginal improvement compared to NFHS-3 (61.5%; 95%CI: 59.4, 63.7%). However, in NFHS-4, 11 states and UTs had stunting more than the national average. These included all the Northeastern states, Jharkhand (47.5%), Bihar (44.1%), Orissa (41.5%), West Bengal (38.8%), Chhattisgarh (38.6%), and Uttar Pradesh (38.2%). These states became worse-off as compared to NFHS-3 in terms of the burden of stunting. In the NFHS-4 were only three states that had a stunting prevalence of less than 20%.

## Discussion

We conducted secondary data analyses of nutritional assessment of adolescents (15–19 years) available from the NFHS-3 and 4. We used the WHO 2007 Growth reference as against the

**Table 7. State level prevalence of thinness (BMI-for-age z-scores <-2SD) and overweight (BMI-for-age z-scores >1 SD) in NFHS-4 (2015–16).**

| Region (N) | % Girls with thinness (CI) | % of boys with thinness (CI) | % thinness both sexes (CI) | % Girls with overweight (CI) | % boys with overweight (CI) | % overweight both sexes (CI) |
|---|---|---|---|---|---|---|
| Andaman (479) | 4.4 (2.4,6.5) | 16 (4.8,27.2) | 5.6 (3.5,7.8) | 12.4 (9.1,15.6) | 18 (6.4,29.6) | 12.9 (9.8,16.1) |
| Andhra Pradesh (1,471) | 10.2 (8.5, 11.9) | 18.3 (12.5, 24.1) | 11.2 (9.6, 12.9) | 9.3 (7.6, 10.9) | 11.8 (6.9, 16.7) | 9.6 (8, 11.1) |
| Arunachal (2,462) | 2 (1.4, 2.6) | 3.3 (1.1, 5.5) | 2.2 (1.6, 2.7) | 8.1 (6.9, 9.3) | 8.6 (5.3, 11.9) | 8.2 (7.1, 9.3) |
| Assam (5,316) | 7.5 (6.7, 8.2) | 8.6 (6.4, 10.9) | 7.6 (6.9, 8.3) | 4.6 (3.9, 5.2) | 5.2 (3.4, 6.9) | 4.6 (4.1, 5.2) |
| Bihar (11,128) | 9.6 (9, 10.2) | 19.6 (17.3, 21.9) | 10.7 (10.1, 11.3) | 2.5 (2.1, 2.8) | 3.2 (2.2, 4.3) | 2.5 (2.2, 2.8) |
| Chandigarh (126) | 9.9 (3.6, 16.2) | 26.9 (8, 45.9) | 13.4 (7.1, 19.7) | 11.9 (5.1, 18.7) | 19.2 (2.2, 36.3) | 13.4 (7.1, 19.7) |
| Chhattisgarh G (5,444) | 8.1 (7.3, 8.9) | 13.5 (10.8, 16.3) | 8.7 (8, 9.5) | 3.3 (2.8, 3.8) | 3.6 (2.1, 5.1) | 3.3 (2.9, 3.8) |
| Dadra Nagarhaveli (170) | 15.6 (9.1, 22) | 14.7 (1.3, 28.1) | 15.4 (9.6, 21.1) | 5.9 (1.6, 10.3) | 11.8 (0, 24.1) | 7.1 (2.9, 11.3) |
| Daman Diu (286) | 11.8 (7.3, 16.4) | 20.3 (10.4, 30.1) | 14 (9.8, 18.2) | 10.9 (6.5, 15.3) | 10.8 (3.1, 18.6) | 10.9 (7.1, 14.7) |
| Delhi (872) | 7.9 (5.9, 9.8) | 11.6 (5.5, 17.7) | 8.4 (6.5, 10.3) | 9.3 (7.2, 11.5) | 8.3 (2.9, 13.6) | 9.2 (7.2, 11.1) |
| Goa (368) | 15.2 (10.5, 19.9) | 10.5 (4.7, 16.3) | 13.6 (10, 17.2) | 9.4 (5.6, 13.3) | 14.5 (7.9, 21.1) | 11.1 (7.8, 14.5) |
| Gujarat (4,714) | 17.8 (16.6, 19.1) | 25.3 (22.5, 28.1) | 19.3 (18.2, 20.5) | 6 (5.2, 6.8) | 8.7 (6.9, 10.6) | 6.5 (5.8, 7.3) |
| Haryana (3,908) | 8 (7, 8.9) | 10.3 (7.7, 12.8) | 8.3 (7.4, 9.2) | 5 (4.2, 5.7) | 6.8 (4.7, 9) | 5.2 (4.5, 6) |
| Himachal (1,791) | 9.8 (8.2, 11.4) | 15.1 (11.5, 18.7) | 11 (9.5, 12.5) | 4.9 (3.7, 6) | 6.3 (3.9, 8.8) | 5.2 (4.1, 6.3) |
| Jammu and Kashmir (5,193) | 5.2 (4.5, 5.9) | 8.8 (7%, 10.5) | 5.9 (5.3, 6.6) | 9 (8.1, 9.8) | 7.2 (5.6, 8.8) | 8.6 (7.8, 9.4) |
| Jharkhand (6,449) | 8.8 (8.1, 9.6) | 17.7 (14.8, 20.6) | 9.8 (9.1, 10.6) | 2.5 (2.1, 3) | 2.7 (1.4, 3.9) | 2.6 (2.2, 3) |
| Karnataka (4,363) | 13.3 (12.2, 14.4) | 24.9 (21.1, 28.6) | 14.8 (13.7, 15.8) | 6 (5.3, 6.8) | 4.6 (2.7, 6.4) | 5.8 (5.1, 6.5) |
| Kerala (1,799) | 7.2 (5.8, 8.5) | 9.8 (6.3, 13.3) | 7.6 (6.4, 8.9) | 9.4 (7.9, 11) | 15 (10.8, 19.1) | 10.4 (9, 11.8) |
| Lakshadweep (143) | 10.2 (4.6, 15.9) | 0 (0, 3.3) | 9.2 (4.1, 14.3) | 11 (5.2, 16.9) | 13.3 (0, 33.9) | 11.3 (5.7, 16.8) |
| Manipur (2,408) | 2.2 (1.5, 2.8) | 3.3 (1.1, 5.5) | 2.3 (1.7, 3) | 8.2 (7, 9.4) | 8.6 (5.3, 12) | 8.2 (7.1, 9.4) |
| Meghalaya (2,049) | 1.5 (0.9, 2.1) | 3.3 (0.8, 5.8) | 1.7 (1.1, 2.3) | 4.9 (3.8, 5.9) | 3.7 (1.1, 6.3) | 4.7 (3.8, 5.7) |
| Maharashtra (5,446) | 13.9 (12.9, 14.9) | 24.4 (21.3, 27.6) | 15.4 (14.4, 16.3) | 6 (5.3, 6.7) | 6.8 (4.9, 8.7) | 6.1 (5.5, 6.8) |
| Mizoram (2,231) | 1.3 (0.8, 1.9) | 2.2 (0.3, 4.1) | 1.4 (0.9, 2) | 8.8 (7.6, 10.1) | 6.9 (3.7, 10.1) | 8.6 (7.4, 9.8) |
| Madhya Pradesh (13,783) | 10.7 (10.1, 11.3) | 24.8 (22.8, 26.8) | 12.5 (12, 13.1) | 3.3 (2.9, 3.6) | 3.6 (2.7, 4.4) | 3.3 (3, 3.6) |
| Nagaland (1,883) | 2.4 (1.6, 3.1) | 3.5 (0.9, 6.1) | 2.5 (1.8, 3.2) | 5.4 (4.3, 6.6) | 3.9 (1.2, 6.7) | 5.3 (4.2, 6.3) |
| Orissa (6,251) | 7.7 (7, 8.5) | 13.6 (10.8, 16.3) | 8.3 (7.7, 9) | 5.4 (4.8, 6) | 6.2 (4.3, 8.2) | 5.5 (4.9, 6.1) |
| Pondicherry (663) | 11.3 (8.6, 14) | 15.7 (7.6, 23.9) | 11.9 (9.4, 14.5) | 11.5 (8.8, 14.2) | 12.4 (5, 19.8) | 11.6 (9.1, 14.1) |
| Punjab (3,171) | 9.6 (8.5, 10.7) | 12.6 (9.5, 15.6) | 10.1 (9, 11.1) | 5.8 (4.9, 6.7) | 10.1 (7.3, 12.9) | 6.4 (5.6, 7.3) |
| Rajasthan (9,521) | 11.8 (11.1, 12.5) | 19.2 (16.9, 21.4) | 12.8 (12.1, 13.4) | 3.1 (2.7, 3.4) | 4.3 (3.1, 5.5) | 3.2 (2.9, 3.6) |
| Sikkim (970) | 2.3 (1.3, 3.4) | 2.6 (0, 5.9) | 2.4 (1.4, 3.4) | 7.9 (6, 9.7) | 17.1 (9.8, 24.3) | 9 (7.1, 10.8) |
| Telangana (1,185) | 15.5 (13.2, 17.7) | 28 (20.9, 35.1) | 17.2 (15, 19.4) | 7.3 (5.6, 8.9) | 6 (2.1, 9.8) | 7.1 (5.6, 8.6) |
| Tamil Nadu (4,772) | 11.2 (10.3, 12.2) | 19.1 (16.2, 21.9) | 12.5 (11.5, 13.4) | 8.1 (7.2, 9) | 8.8 (6.7, 10.9) | 8.2 (7.4, 9) |
| Tripura (849) | 4.5 (2.9, 6.1) | 12.3 (5.8, 18.7) | 5.5 (3.9, 7.1) | 7.6 (5.6, 9.6) | 7.9 (2.5, 13.3) | 7.7 (5.8, 9.5) |
| Uttarakhand (3,707) | 6.8 (5.9, 7.7) | 13.2 (10, 16.5) | 7.6 (6.7, 8.5) | 5.8 (5%, 6.6) | 6.4 (4, 8.8) | 5.9 (5.1, 6.6) |
| Uttar Pradesh (25,554) | 8.3 (7.9, 8.6) | 17.4 (16, 18.7) | 9.3 (9, 9.7) | 3.8 (3.6, 4.1) | 3.9 (3.2, 4.6) | 3.8 (3.6, 4.1) |
| West Bengal (3,448) | 9 (8, 10) | 16.4 (12.5, 20.3) | 9.8 (8.8, 10.8) | 6 (5.1, 6.8) | 4.5 (2.3, 6.7) | 5.8 (5, 6.6) |

Thinness, severe thinness, overweight in adolescents are based on the 2007 WHO Growth Reference. Overweight percentages include obesity.

adult cut-offs that are presently being used for adolescents in these surveys. Due to possible non-availability of this highly mobile age-group and removal of improbable values outside the default upper and lower boundaries, the anthropometric data available for final analysis was 68% in NFHS-3 (48.4% boys; 87.4% girls) and 52% in NFHS-4 (13.4% boys; 92.9% girls).

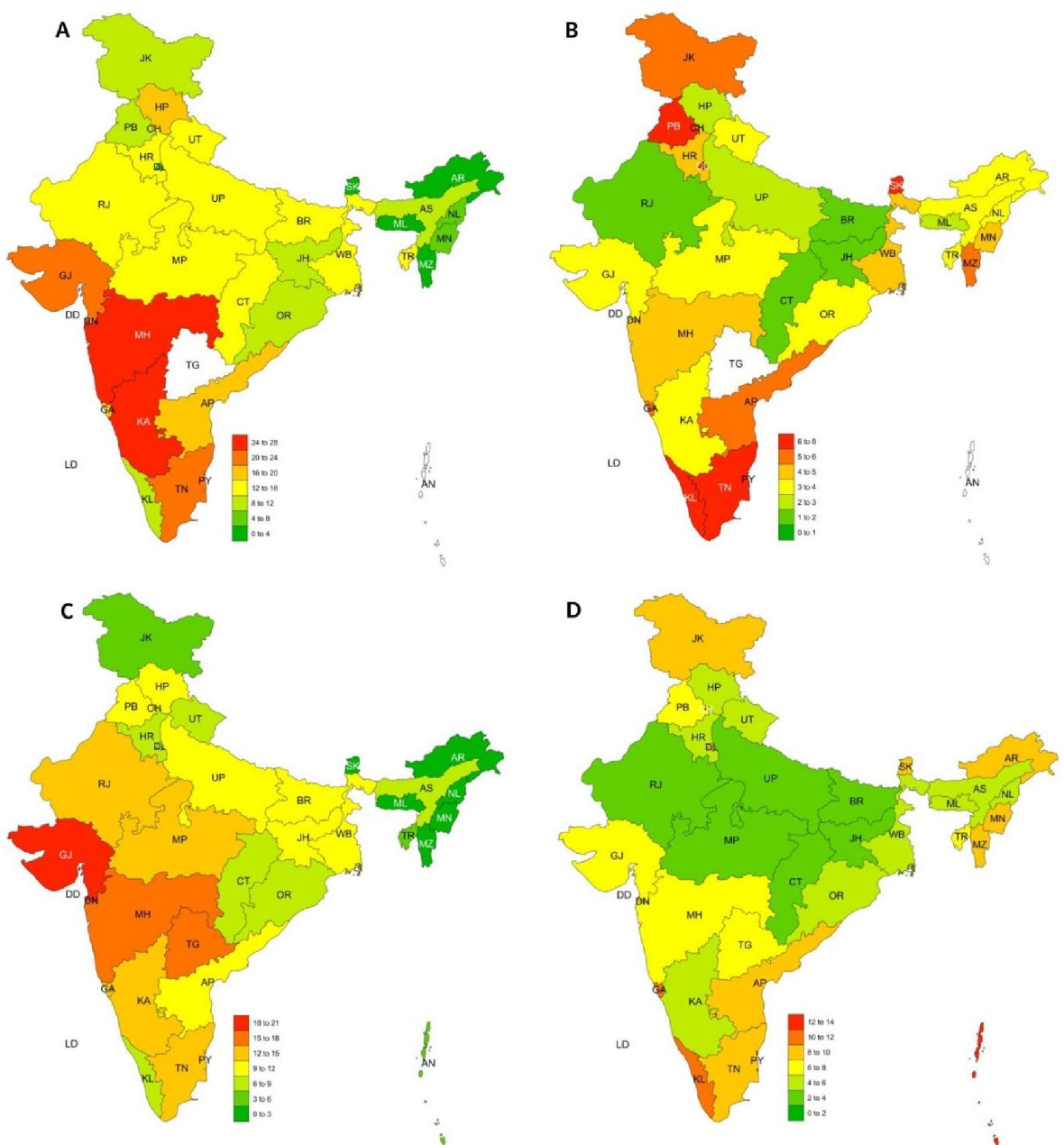

**Fig 5.** Prevalence of thinness (A and C) and overweight (B and D) in adolescents (15–19 years) in NFHS-3 (A and B) NFHS-4 (C and D) using WHO 2007 Growth Reference.

We found that the thinness in boys and girls was 22.3% and 9.9% in NFHS-3 and 16.5% and 9% in NFHS-4. The prevalence of stunting was 25.2% and 34.1% in NFHS-3; 32.2% and 34.4% in NFHS-4 in boys and girls respectively. These revised estimates indicate a dramatically different adolescent nutrition status in India as against what is currently reported. The problem of stunting in this age group is much bigger than thinness and there is a trend indicating an increase in overweight.

**Table 8. State level prevalence of stunting (height-for-age z-scores <-2SD) in adolescents (15–19 years) by state in NFHS-3 (2005–6).**

| Region (N) | % Girls with stunting (CI) | % of boys with stunting (CI) | % stunting both sexes (CI) |
|---|---|---|---|
| Andhra Pradesh (2,384) | 28.5 (26%, 31.1%) | 19.3 (16.9%, 21.6%) | 24.2 (22.5%, 26%) |
| Arunachal Pradesh (558) | 44.8 (39.8%, 49.8%) | 40.9 (32.8%, 49%) | 43.7 (39.5%, 47.9%) |
| Assam (886) | 42.6 (38.7%, 46.4%) | 30.5 (24.3%, 36.6%) | 39.4 (36.1%, 42.7%) |
| Bihar (1,070) | 34.1 (30.8%, 37.5%) | 23.2 (17.6%, 28.8%) | 37.6 (34.6%, 40.5%) |
| Chhattisgarh (1,053) | 34.1 (30.8%, 37.5%) | 26.3 (20.7%, 31.8%) | 32.2 (29.3%, 35.1%) |
| Delhi (680) | 30.9 (26.6%, 35.2%) | 27.4 (21.1%, 33.6%) | 29.8 (26.3%, 33.3%) |
| Goa (648) | 27.5 (23.5%, 31.5%) | 24.7 (17.4%, 31.9%) | 26.9 (23.4%, 30.3%) |
| Gujarat (910) | 24.1 (20.8%, 27.4%) | 21.1 (15.7%, 26.5%) | 23.3 (20.5%, 26.1%) |
| Haryana (759) | 15.6 (12.4%, 18.7%) | 13.1 (8.4%, 17.9%) | 14.9 (12.3%, 17.5%) |
| Himachal (730) | 19.2 (15.8%, 22.7%) | 14.9 (9.7%, 20%) | 18 (15.2%, 20.9%) |
| Jammu and Kashmir (897) | 16.9 (13.9%, 19.9%) | 17.1 (12.3%, 22%) | 17 (14.5%, 19.5%) |
| Jharkhand (768) | 47.9 (43.7%, 52%) | 31.4 (24.4%, 38.3%) | 43.9 (40.3%, 47.5%) |
| Karnataka (1,810) | 26.6 (23.8%, 29.3%) | 26.6 (23.4%, 29.7%) | 26.6 (24.5%, 28.6%) |
| Kerala (705) | 22 (18.4%, 25.6%) | 11.9 (6.7%, 17.1%) | 19.6 (16.6%, 22.6%) |
| Manipur (1,381) | 31.1 (27.7%, 34.5%) | 26.3 (22.8%, 29.9%) | 28.9 (26.5%, 31.4%) |
| Meghalaya (596) | 64.3 (59.7%, 68.8%) | 62.9 (54.9%, 70.9%) | 63.9 (60%, 67.9%) |
| Maharashtra (2,856) | 29.7 (27.4%, 32.1%) | 21.1 (18.9%, 23.2%) | 25.5 (23.9%, 27.1%) |
| Mizoram (426) | 33 (27.6%, 38.4%) | 35 (26%, 44.1%) | 33.6 (29%, 38.2%) |
| Madhya Pradesh (1,787) | 27.5 (25%, 30%) | 25.2 (21.2%, 29.1%) | 26.9 (24.8%, 28.9%) |
| Nagaland (1,515) | 27.6 (24.4%, 30.8%) | 41.1 (37.4%, 44.7%) | 34.1 (31.6%, 36.5%) |
| Orissa (6,251) | 37.7 (34.4%, 41%) | 24.5 (18.7%, 30.2%) | 34.8 (32%, 37.7%) |
| Punjab (888) | 13.8 (11%, 16.6%) | 16.7 (11.9%, 21.4%) | 14.6 (12.3%, 17%) |
| Rajasthan (1,135) | 18.1 (15.4%, 20.7%) | 14.2 (10%, 18.3%) | 17.1 (14.8%, 19.3%) |
| Sikkim (594) | 41.9 (37.2%, 46.5%) | 52.8 (44.3%, 61.3%) | 44.5 (40.4%, 48.6%) |
| Tamil Nadu (1,656) | 24.6 (21.7%, 27.5%) | 22.3 (19.3%, 25.2%) | 23.5 (21.4%, 25.6%) |
| Tripura (566) | 45 (40.3%, 49.8%) | 40.2 (31.1%, 49.3%) | 44 (39.8%, 48.2%) |
| Uttarakhand (778) | 25.9 (22.3%, 29.5%) | 27.3 (20.4%, 34.1%) | 26.2 (23.1%, 29.4%) |
| Uttar Pradesh (4,822) | 35.5 (33.6%, 37.5%) | 24 (22.3%, 25.8%) | 29.9 (28.6%, 31.2%) |
| West Bengal (1,632) | 38.9 (36.2%, 41.7%) | 29.7 (25%, 34.4%) | 36.8 (34.4%, 39.1%) |

Stunting in adolescents is based on the 2007 WHO Growth Reference.

## Overestimation of thinness when adult cut-offs are used for adolescents

The use of adult cut-offs rather than the recommended WHO 2007 Growth Reference seems to overestimate the thinness by almost 2.5 fold in the boys and 4 fold in the girls [11,12]. These are also more consistent and plausible with the trends in the older age groups (20–29, 30–39, 40–49 years) as seen in Fig 7. Moreover, the over-estimation in one particular age-group (15–19 years) is likely to inflate the overall prevalence of thinness in adults and may not give an actual picture of the nutrition transition and evolving double burden of malnutrition in the country.

The results of our re-analysis are comparable to those in other studies using the WHO 2007 Growth Reference. According to the Non-communicable Disease Risk Factor Collaboration (NCD-RisC), a database of 2416 population-based studies on cardio-metabolic risk factors, thinness in the age-group 5–19 years was 30.7% (95% CI: 23.5, 38.0) in boys and 22.7% (95% CI 16.7, 29.6) in girls for India [17]. These are higher but closer to our re-estimation than the currently reported thinness by the NFHS. In the Global School-Based survey, thinness in the

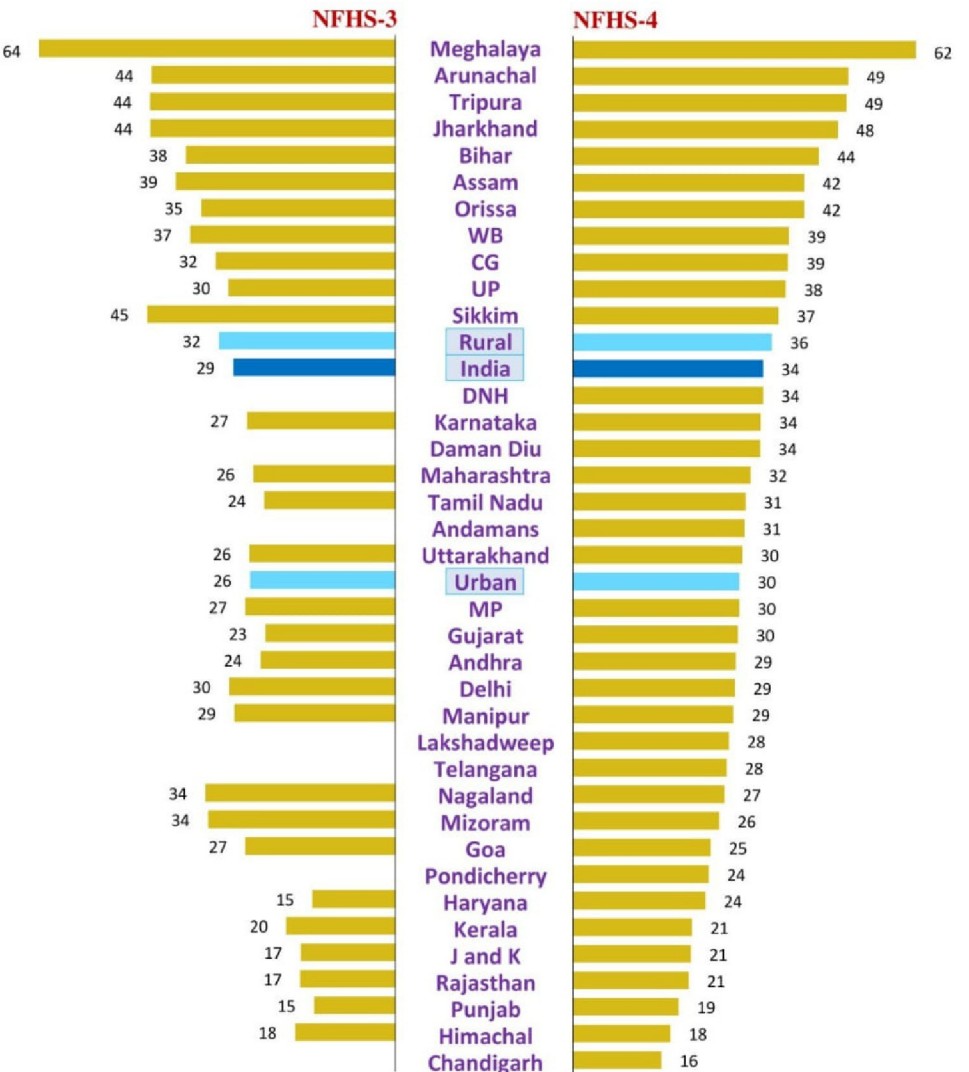

**Fig 6. Prevalence of stunting in adolescents (15–19 years) by states/union territories in NFHS-3 and 4.**

age group 12–15 years was 17.2% (95%CI: 14.2, 20.8) in boys and 14.1% (95%CI: 11.4, 17.3) in girls in India using the WHO 2007 growth reference [18]. A recent study from Orissa and Chhattisgarh found thinness to be 9.6% in girls in 10–19 years age-group using BAZ and mid-upper arm circumference [19].Overall in South Asia, the prevalence of thinness has been higher in boys (28.6%) as compared to girls (20.3%) [17]. Among countries with comparable nutritional indicators, thinness was 32.4% in boys and 21.4% in girls (10–19 years) in Ethiopia,11.9% in girls (12–19 years) in Bangladesh, and 15.2% in Somalian refugee girls (10–19 years) [20–23].

## Stunting estimates in adolescents and the trend over NFHS-3 and NFHS-4

Stunting is a robust indicator of long-term nutrition at the population level. In the NFHS the estimates of stunting are limited to the under-five children and no stunting is ever reported in any age-group other than the under-five at the national level.

**Table 9. State level prevalence of stunting (height-for-age z-scores <-2SD) in adolescents (15–19 years) by state in NFHS-4 (2015–16).**

| Region (N) | % Girls with stunting (CI) | % Boys with stunting (CI) | % stunting both sexes (CI) |
|---|---|---|---|
| Andamans (479) | 29.5 (25.1%, 34) | 41.2 (26.7%, 55.7%) | 30.8 (26.5%, 35%) |
| Andhra Pradesh (1,471) | 30.2 (27.6%, 32.7%) | 22 (15.8%, 28.3%) | 29.2 (26.8%, 31.5%) |
| Arunachal Pradesh (2,462) | 48.5 (46.4%, 50.6%) | 56.1 (50.4%, 61.9%) | 49.4 (47.4%, 51.4%) |
| Assam (5,316) | 42.1 (40.7%, 43.5%) | 37.3 (33.5%, 41%) | 41.5 (40.2%, 42.9%) |
| Bihar (11,128) | 44.9 (43.9%, 45.9%) | 37.2 (34.5%, 40%) | 44.1 (43.2%, 45%) |
| Chandigarh (126) | 16 (8.3%, 23.7%) | 15.4 (0%, 31.2%) | 15.9 (9.1%, 22.7%) |
| Chhattisgarh (5,444) | 38.1 (36.7%, 39.5%) | 41.9 (38%, 45.7%) | 38.6 (37.3%, 39.9%) |
| Dadra Nagarhaveli (170) | 31.6 (23.4%, 39.8%) | 44.1 (26%, 62.3%) | 34.1 (26.7%, 41.5%) |
| Daman Diu (286) | 32.1 (25.6%, 38.6%) | 37.8 (26.1%, 49.6%) | 33.6 (27.9%, 39.2%) |
| Delhi (872) | 28.6 (25.3%, 31.9%) | 32.2 (23.5%, 41%) | 29.1 (26.1%, 32.2%) |
| Goa (368) | 27 (21.3%, 32.8%) | 20.2 (12.7%, 27.6%) | 24.7 (20.2%, 29.3%) |
| Gujarat (4,714) | 30 (28.5%, 31.5%) | 27.8 (24.8%, 30.7%) | 29.6 (28.3%, 30.9%) |
| Haryana (3,908) | 23 (21.5%, 24.4%) | 28.5 (24.8%, 32.3%) | 23.8 (22.4%, 25.1%) |
| Himachal (1,791) | 18 (15.9%, 20%) | 15.8 (12.1%, 19.4%) | 17.5 (15.7%, 19.3%) |
| Jammu and Kashmir (5,193) | 20.4 (19.1%, 21.6%) | 24.3 (21.7%, 27%) | 21.2 (20%, 22.3%) |
| Jharkhand (6,449) | 44.2 (40.5%, 47.9%) | 47.9 (46.6%, 49.2%) | 47.5 (46.3%, 48.7%) |
| Karnataka (4,363) | 32.8 (31.3%, 34.3%) | 40 (35.8%, 44.2%) | 33.7 (32.3%, 35.1%) |
| Kerala (1,799) | 19.9 (17.8%, 22%) | 28.8 (23.5%, 34%) | 21.4 (19.5%, 23.3%) |
| Lakshadweep (143) | 25 (17.1%, 32.9%) | 53.3 (24.8%, 81.9%) | 28 (20.3%, 35.7%) |
| Manipur (2,408) | 28.8 (26.8%, 30.7%) | 29.2 (23.9%, 34.5%) | 28.8 (27%, 30.7%) |
| Meghalaya (2,049) | 61.6 (59.4%, 63.9%) | 60.7 (54.4%, 67.1%) | 61.5 (59.4%, 63.7%) |
| Maharashtra (5,446) | 31.4 (30.1%, 32.7%) | 35.1 (31.6%, 38.5%) | 31.9 (30.7%, 33.2%) |
| Mizoram (2,231) | 25.9 (24%, 27.9%) | 29.1 (23.5%, 34.6%) | 26.3 (24.5%, 28.2%) |
| Madhya Pradesh (13,783) | 30 (29.1%, 30.8%) | 29.2 (27.1%, 31.4%) | 29.9 (29.1%, 30.6%) |
| Nagaland (1,883) | 25.5 (23.4%, 27.6%) | 39.5 (32.9%, 46%) | 27.2 (25.2%, 29.2%) |
| Orissa (6,251) | 41.8 (40.5%, 43.1%) | 39.1 (35.2%, 42.9%) | 41.5 (40.3%, 42.8%) |
| Pondicherry (663) | 22.8 (19.3%, 26.3%) | 34.8 (24.4%, 45.3%) | 24.4 (21.1%, 27.8%) |
| Punjab (3,171) | 18.4 (16.9%, 19.9%) | 22.5 (18.7%, 26.3%) | 19 (17.7%, 20.4%) |
| Rajasthan (9,521) | 25.5 (23%, 28%) | 20.1 (19.3%, 21%) | 20.8 (20%, 21.6%) |
| Sikkim (970) | 33.5 (30.3%, 36.8%) | 61.5 (52.3%, 70.8%) | 36.9 (33.8%, 40%) |
| Telangana (1,185) | 29.7 (26.8%, 32.6%) | 14.9 (9.2%, 20.6%) | 27.6 (25%, 30.2%) |
| Tamil Nadu (4,772) | 30.2 (28.8%, 31.7%) | 34.8 (31.4%, 38.3%) | 31 (29.6%, 32.3%) |
| Tripura (849) | 49.2 (45.5%, 52.9%) | 48.7 (39%, 58.3%) | 49.1 (45.7%, 52.5%) |
| Uttarakhand (3,707) | 27.9 (23.6%, 32.1%) | 30.8 (29.2%, 32.4%) | 30.4 (28.9%, 31.9%) |
| Uttar Pradesh (25,554) | 39.4 (38.8%, 40%) | 29.6 (27.9%, 31.2%) | 38.2 (37.6%, 38.8%) |
| West Bengal (3,448) | 39.8 (38.1%, 41.6%) | 30.7 (25.9%, 35.5%) | 38.8 (37.2%, 40.4%) |

Stunting in adolescents is based on the 2007 WHO Growth Reference.

Our analysis for the first time offers the estimates of stunting in adolescents in nationally representative samples. The overall prevalence of stunting in adolescents was 29.1% (25.2% in boys, 31.2% in girls) in NFHS-3 which increased to 34.1% in NFHS-4 (32.2% in boys and 34.4% in girls). The increase in stunting is supported by the fact that there were six states in NFHS-3 that had stunting less than 20%, whereas there were only three such states in NFHS-4. This may be either a true increase but is confounded by the differing nature of populations sampled in both the surveys. The NFHS-4 sample was larger and had a greater representation of households from rural areas from poorer wealth quintiles. There was also a great

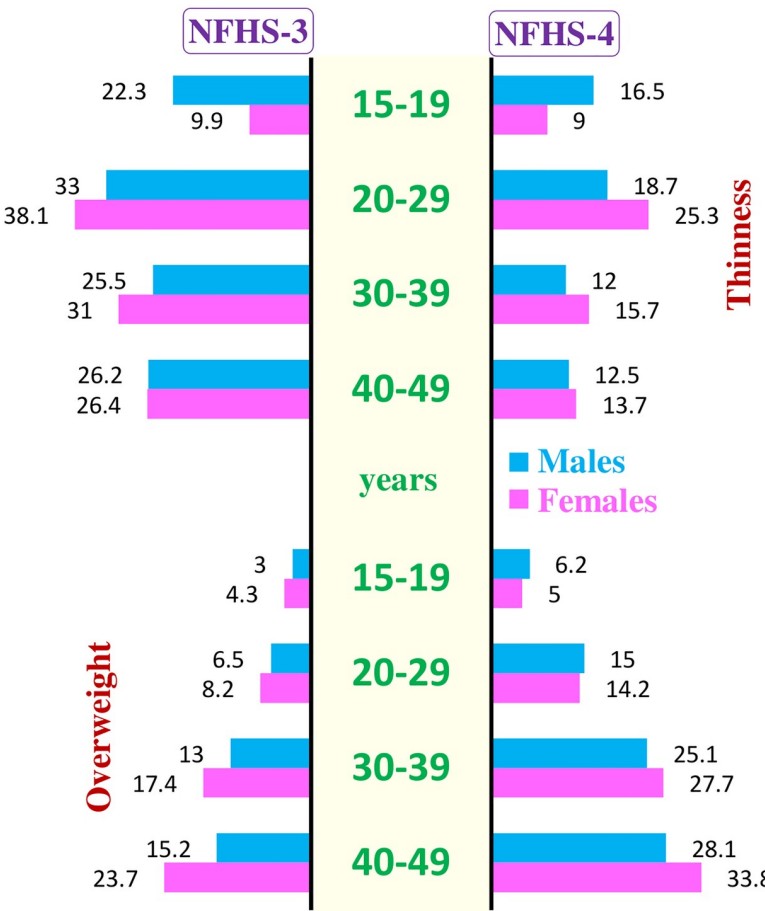

**Fig 7. Comparison of thinness and overweight in late adolescents (15–19 years) and adults (20–49) in NFHS-3 and 4.**

subnational heterogeneity varying from 63.9% (NFHS-3) and 61.5% (NFHS-4) in Meghalaya to 14.6% in Punjab in the NFHS-3 and 15.9% in Chandigarh in the NFHS-4.

There is a paucity of data on adolescent/adult stunting in India. The National Nutritional Monitoring Bureau (NNMB) conducted a study comprising of diet and nutritional assessment in nine states in 2006 [24]. It reported increasing levels of stunting across childhood and adolescence. The prevalence of stunting was 30.1% in the 6–9 year age group, 34.2% in 10–13 and 36.8% in 14-17-year-olds, with no significant difference between sexes at any age. According to the Global School-Based survey, it was 14.2% (95%CI: 10.6, 18.8) in boys and 15.1% (95% CI: 9.9, 22.4) in girls in India, but this was in the age-group 12–15 years [18]. Levels of stunting reported in other studies have been higher in Asia; 48% in adolescents from rural Bangladesh [25] and 23.7% in Malaysia [26], compared to 12.2% in Ethiopian adolescent girls [20] and 9.7% in Somalian refugee camps [23].

## Determinants of stunting in adolescents

The height attained in adulthood is a combination of pre-pubertal height and height attained during the pubertal growth spurt. India has a huge (46.6 million) population of under-five children with stunting [27]. The high levels of stunting in Indian adolescents could represent the persistence of pre-pubertal deficit. The pubertal growth spurt which can add 8.3 cm/year in

girls and 9.5 cm/year in boys requires additional nutrition [28]. In countries with high levels of malnutrition, the adolescent growth period may be prolonged in duration and may continue till the late adolescence [29]. The NNMB study documented food intakes in 10-17-year-old adolescents. The median caloric intake in boys ranged from 1387 KCalories in the 10–12 years, 1611 in the 13–15 years, and 1832 in 16–17 year age groups. Similarly, protein intakes were 36 gm, 42 gm, and 50 gm in these age groups respectively [24]. These are substantially below the recommended allowance and the intakes in the girls were even lower.

Stunting in our analysis was worst in girls and adolescents from the rural areas and the poorer quintiles, reflecting the impact of gender, place of residence, and socioeconomic status on long-term nutrition. Stunting is an indicator that is affected by food security, education, water sanitation and hygiene, disparity, economic development, and women's health and empowerment [30]. Stunting was similar in both the sexes in the under-five age group in both the NFHS, but the girls ended up being more stunted during adolescence as per our analysis [11,12]. This was also demonstrated in a review of the NNMB data (children 0–18 years over two surveys conducted in 1975–9 and 2012–13 [31]. The boys and girls had similar growth faltering compared to the WHO median till the age of 14 years but the boys grew better after that [31].

## Implications of stunting in adolescents

Stunting in adolescents has received less attention as a public health problem in India, since there has been no nationally representative survey of adolescents. It also goes unrecognized in communities where it is common and gets normalized [30]. The first 1000 days are critical for growth and development in infants due to maximum growth velocity in early life [30,32]. It is however also recognized that the next 7000 days of childhood to adulthood are equally important for growth [33]. The growth velocity at puberty is similar to that in the first two years of life. While the brain attains 95% of its adult size by 6 years of age, memory, emotional processing, decision making, and higher executive functions develop during mid-childhood and adolescence [33]. An increase in height by one SD was associated with 5% points more likelihood of being able to write, indicating its role in cognitive development [34].There is a potential for catch up growth in adolescence in countries like India, where growth may continue to falter beyond the first 1000 days and it is an additional 'window of opportunity' for positive intervention and catch-up [29]. Its importance in adolescent girls is even more important given the inter-generational consequences [35]. The persistently high levels of stunting in Indian adolescents indicate a lost opportunity for normal growth and development, including cognitive development.

## Stunting and its impact on estimates of thinness

Stunting and thinness are both indices of undernutrition in adolescents and both use the measurement of height. While lower HAZ score indicates chronic undernutrition, a low BAZ score is indicative of acute undernutrition. However stunting impacts the estimates of BAZ leading to underestimation of thinness, the mean BAZ may appear better in stunted populations than that might be in a population with better heights. This might be one of the reasons for low levels of thinness seen in our analysis, especially in the girls. The phenomenon where boys continue to gain height resulting in higher levels of thinness has also been seen in neighboring countries like Indonesia, Bangladesh and Nepal (11%, 19.6%, 37.8% in boys and 5%, 15.4% and 26.2% in girls) [36–38].

## Over-nutrition using WHO 2007 Growth Reference in adolescents

The impact of the use of WHO 2007 Growth Reference on the prevalence of overweight in adolescents was modest, and more so for obesity. There is a trend of an increase in overweight in this age-group between the two surveys. While no state had overweight prevalence greater than 10% in the NFHS-3, there were seven such states and UTs in the NFHS-4. It continued to be less than 4% in states with a higher prevalence of thinness like Madhya Pradesh, Chhattisgarh, Rajasthan, Jharkhand, and Bihar.

A comparison with other studies needs to be done cautiously due to the use of different classifications and cut-offs for overweight/obesity. Studies have variously used the cut-offs recommended by the International Obesity Task Force (IOTF), CDC and occasionally national standards [39–42]. In subnational studies in India, the prevalence of overweight/obesity in adolescents was 7.8% in urban Gujarat and 15.6% in Uttarakhand using WHO growth reference [43, 44]; 21% in urban Bihar using the CDC cut-offs [45]; 11% in urban Haryana [46] and 7.2% in Telangana using IOTF cut-offs [47].A large multi-centric study in affluent school going children (2–17 years) in India estimated the overall prevalence of overweight/obesity as 18.2% by IOTF classification and 23.9% by the WHO standards [48]. A systematic analysis of 1,769 global studies found the prevalence of overweight in children and adolescents in developing countries to have increased in boys from 8.1% (95%CI: 7.7, 8.6) to 12.9% (95%CI: 12.3, 13.5)in boys and from 8.4% (95%CI: 8.1, 8.8) to 13.4% (95%CI: 13.0, 13.9) in girls over 1980 to 2013 [49].

## Double-burden of malnutrition

The WHO defines the double burden of malnutrition as the coexistence of undernutrition along with overweight, obesity or diet-related NCDs, within individuals, households, and populations and across the life-course [50]. In operational and statistical terms many surveys have defined double burden as the high prevalence of wasting, stunting or thinness as well as overweight in any population group. The cut-offs used to define the high prevalence of undernutrition are, for example, wasting >15%, thinness > 20%, stunting > 30%, and overweight in children or adults variously as more than 20%, 30%, or 40% [51]. India too is reported as having a double burden with a high prevalence of undernutrition in adults and children; emerging problems of overnutrition and NCDs, especially in the urban areas and high prevalence of micronutrient deficiency [52]. Our reanalysis of NFHS-3 and 4 show that there are increasing levels of overweight in some states along with the persistence of thinness and stunting in others. In states like Kerala, the persistence of thinness and stunting alongside overweight do suggest loci of a double burden of malnutrition. These are influenced by age, gender, residence, region, and income. In a study from urban Hyderabad (12–17 year), overweight was four times higher in the upper socio-economic class (OR: 4.1; 95%CI: 2.25,7.52) [47].A cross-sectional analysis of 57 DHS across low and middle-income countries conducted in 1994–2008 did not find a substantial co-existence of under and over-nutrition [53]. Moreover, with the high prevalence of stunting in the age-group which is at the threshold of adulthood, the prevalence of thinness needs interpretation with caution.

Poor growth and height are closely associated with poverty and deprivation, reflected in the phrase by Tanner, 'growth is a mirror of the conditions of society' [54]. In the present analysis, we found that stunting prevalence was either equal or higher in boys in the better performing states and it was more in girls in the worst-performing states. The Empowered Action Group states that are performing poorly in other social, economic and health indicators have higher stunting which has worsened from NFHS-3 to NFHS-4and this is corroborated by others [55–57].

## Strengths and limitations

Our re-analysis used the recommended WHO 2007 Growth Reference for the characterization of the nutritional status of adolescents in successive editions of the NFHS across a span of 10 years. This is the first snapshot of the nutritional status of adolescents in nationally representative samples. This is an addition to the sparse literature in adolescent nutrition in India and abroad. According to a bibliometric analysis, only 1.2% of the publications were pertaining to adolescents as against more than 95% for the under-five [33].

However, there are several limitations to be considered in the interpretation of the findings. First, the NFHS covered only boys and girls in the late adolescence with no representation of 10–14 years age group. Also, anthropometry was available in a limited proportion of adolescents, especially boys. Second, the possibility of inaccurate age reporting cannot be ruled out. Third, the WHO 2007 Growth Reference has limitations as these do not account for the complex racial/ethnic variation across populations in the timing of the adolescent growth spurt and are not based on current or future health risks in them [58].

## Conclusions

The revised national and sub-national estimates of thinness, stunting and overweight in the age-group of 15–19 years using the WHO 2007 Growth reference reveal several novel findings relevant for adolescent and adult health in India. The prevalence of thinness is 2.5–4 fold lower in boys and girls than what has been reported using the adult cutoffs in both NFHS-3 and NFHS-4. The prevalence of thinness is higher in the boys and that of stunting is higher in the girls. Stunting appears to have increased over the two surveys and affects one in three adolescents. It indicates a lost opportunity for undoing the damage of stunting in childhood and the potential for nutritional recovery in the second decade. Adolescents in rural India, living in poverty and the less developed states continue to suffer nutritional deprivation which has implications for their growth, development, predisposition to communicable and non-communicable diseases, and their escape from poverty. There is a need for nutritional interventions for improving adolescent nutrition in India which receives scant attention at the moment. These findings have future implications (NFHS-5) so that vital information on the nutritional status of adolescents and nutrition transition in the population is neither buried in the adult data nor does it influence adult means inappropriately.

## Supporting information

**S1 Data. Mean height-for-age and weight-for-age z-scores of all states in NFHS-3 and 4.** (XLSX)

## Acknowledgments

Authors gratefully acknowledge comments and inputs of Dr. Anura Kurpad, Division of Nutrition, St John's Research Institute, Bangalore.

## Author Contributions

**Conceptualization:** Madhavi Bhargava, Anurag Bhargava.

**Data curation:** Madhavi Bhargava, Anurag Bhargava, Sudeep D. Ghate, R. Shyama Prasad Rao.

**Formal analysis:** Madhavi Bhargava, Anurag Bhargava, R. Shyama Prasad Rao.

**Funding acquisition:** Madhavi Bhargava.

**Methodology:** Madhavi Bhargava, Anurag Bhargava, Sudeep D. Ghate, R. Shyama Prasad Rao.

**Project administration:** Madhavi Bhargava.

**Software:** Madhavi Bhargava, R. Shyama Prasad Rao.

**Validation:** Sudeep D. Ghate.

**Visualization:** Madhavi Bhargava, Sudeep D. Ghate, R. Shyama Prasad Rao.

**Writing – original draft:** Madhavi Bhargava.

**Writing – review & editing:** Madhavi Bhargava, Anurag Bhargava, Sudeep D. Ghate, R. Shyama Prasad Rao.

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
