## [Decision Letter · Decision Letter 0]

14 May 2020

PONE-D-20-09042

Nutritional status of Indian adolescents (15-19 years) from National Family Health Surveys 3 and 4: Revised estimates using WHO 2007 Growth reference

PLOS ONE

Dear Dr. Bhargava,

Thank you for submitting your manuscript to PLOS ONE. After careful consideration, we feel that it has merit but does not fully meet PLOS ONE’s publication criteria as it currently stands. Therefore, we invite you to submit a revised version of the manuscript that addresses the points raised during the review process.

I think your manuscript presents very valuable information and argues for using age specific growth charts to assess the nutritional status of adolescents. Both the reviewers have given very favourable comments and have recommended accepting the manuscript. In addition to a minor edit suggested by one of the reviewers, i would also request you to consider editing the word limit of your manuscript, in order to make it more sharp and improve its readability. 

We would appreciate receiving your revised manuscript by Jun 28 2020 11:59PM. To enhance the reproducibility of your results, we recommend that if applicable you deposit your laboratory protocols in protocols.io, where a protocol can be assigned its own identifier (DOI) such that it can be cited independently in the future. For instructions see: http://journals.plos.org/plosone/s/submission-guidelines#loc-laboratory-protocols

We look forward to receiving your revised manuscript.

Kind regards,

Vijayaprasad Gopichandran

Academic Editor

PLOS ONE

Journal Requirements:

2. Thank you for stating the following in your Competing Interests section: 'None'

a. Please complete your Competing Interests statement to state any Competing Interests. If you have no competing interests, please state "The authors have declared that no competing interests exist.", as detailed online in our guide for authors at http://journals.plos.org/plosone/s/submit-now

Reviewers' comments:

Reviewer's Responses to Questions

**Comments to the Author**

1. Is the manuscript technically sound, and do the data support the conclusions?

Reviewer #1: Yes

Reviewer #2: Yes

2. Has the statistical analysis been performed appropriately and rigorously? 

Reviewer #1: Yes

Reviewer #2: Yes

3. Have the authors made all data underlying the findings in their manuscript fully available?

Reviewer #1: Yes

Reviewer #2: Yes

4. Is the manuscript presented in an intelligible fashion and written in standard English?

Reviewer #1: Yes

Reviewer #2: Yes

5. Review Comments to the Author

Reviewer #1: The manuscript is very well written and fulfils the gap in anthropometric data analysis for adolescents in NFHS - 3 and NFHS - 4. All tables and graphs have been presented comprehensively and does complete justice for analysis of demographic data.

Reviewer #2: The manuscript is well written and logically presented. It throws light on the nutritional status of the adolescent age group that is often sidelined. The manuscript also has shown the right method of classification of nutritional status and adolescent age group with use of appropriate standards for assessment of nutritional status. Suggestion to the authors: will there be a difference in classification of nutritional status of the adolescent if IAP classification is used instead of WHO?

6. PLOS authors have the option to publish the peer review history of their article (what does this mean?). If published, this will include your full peer review and any attached files.

Reviewer #1: No

Reviewer #2: No

---

## [Author Response · Author response to Decision Letter 0]

25 May 2020

Reviewer #1: The manuscript is very well written and fulfils the gap in anthropometric data analysis for adolescents in NFHS - 3 and NFHS - 4. All tables and graphs have been presented comprehensively and does complete justice for analysis of demographic data.

RESPONSE: 

We thank you for your encouraging comments.

Reviewer #2: The manuscript is well written and logically presented. It throws light on the nutritional status of the adolescent age group that is often sidelined. The manuscript also has shown the right method of classification of nutritional status and adolescent age group with use of appropriate standards for assessment of nutritional status. 

RESPONSE:

Thank you for your positive comments.

Reviewer #2: Suggestion to the authors: will there be a difference in classification of nutritional status of the adolescent if IAP classification is used instead of WHO?

RESPONSE:

Authors agree that older children and adolescents have growth patterns which may be different in different populations of the world. This is because nutritional, environmental, genetic factors and timing of puberty seem to play a major role not only in the attainment of final height but also in the characteristics of the growth curve. 

It might be a good idea to analyze this data using the IAP charts also to see the difference in estimates, but it is beyond the scope and objective of this manuscript. 

While the IAP charts are very useful in individual surveys, presently there is no software/algorithm available based on the IAP charts that can analyze big data like that in NFHS. But the IAP estimates would definitely be closer to our WHO estimates than those using the adult cut-offs (presently reported by the NFHS). The advantage of using WHO growth reference is that it avails comparability between countries.

Authors thank you for this valuable suggestion and we may think of doing this exercise as a new project.

---

## [Editor Report · Decision Letter 1]

29 May 2020

Nutritional status of Indian adolescents (15-19 years) from National Family Health Surveys 3 and 4: Revised estimates using WHO 2007 Growth reference

PONE-D-20-09042R1

Dear Dr. Bhargava,

We are pleased to inform you that your manuscript has been judged scientifically suitable for publication and will be formally accepted for publication once it complies with all outstanding technical requirements.

With kind regards,

Vijayaprasad Gopichandran

Academic Editor

PLOS ONE
---

## [Editor Report · Acceptance letter]

2 Jun 2020

PONE-D-20-09042R1 

Nutritional status of Indian adolescents (15-19 years) from National Family Health Surveys 3 and 4: Revised estimates using WHO 2007 Growth reference 

Dear Dr. Bhargava:

I'm pleased to inform you that your manuscript has been deemed suitable for publication in PLOS ONE. Congratulations! Your manuscript is now with our production department. 

Kind regards, 

on behalf of

Dr. Vijayaprasad Gopichandran 

Academic Editor

PLOS ONE